RNA sequencing identifies lung cancer lineage and facilitates drug repositioning

Zeng Longjin 1
Zhang Longyao 2
Li Lingchen 2
Liao Xingyun 3
Yin Chenrui 2
Zhang Lincheng 1
Chen Xiewan 1
Sun Jianguo 2 sunjianguo@tmmu.edu.cn
1 Department of Basic Medicine, Army Medical University , Chongqing , China
2 Cancer Institute, Xinqiao Hospital , Chongqing , China
3 Affiliated Tumor Hospital, Department of Oncology , Chongqing , China
Oliveira Sonia
Electronic publication date: 2024 Sep 24
Publication date: 2024
Volume: 12
Electronic Location ID: e18159
Received 2024 May 27; Accepted 2024 Sep 2
Copyright: © 2024 Zeng et al.
Copyright year: 2024
Copyright holder: Zeng et al.
License: This is an open access article distributed under the terms of the Creative Commons Attribution License, which permits unrestricted use, distribution, reproduction and adaptation in any medium and for any purpose provided that it is properly attributed. For attribution, the original author(s), title, publication source (PeerJ) and either DOI or URL of the article must be cited.
License URL: https://creativecommons.org/licenses/by/4.0/

Keywords: Lung adenocarcinoma, Metagene, Molecular classification, Drug sensitivity

Funding: National Natural Science Foundation of China 82473261, 82172670, 81972858, and 82202951 Technology Innovation and Application Development Project of Chongqing 2023DBXM002 and CSTB2022TIAD-KPX0176 This study was supported by the National Natural Science Foundation of China (82473261, 82172670, 81972858, and 82202951), the Technology Innovation and Application Development Project of Chongqing (2023DBXM002 and CSTB2022TIAD-KPX0176). The funders had no role in study design, data collection and analysis, decision to publish, or preparation of the manuscript.

==============================
Recent breakthrough therapies have improved survival rates in non-small cell lung cancer (NSCLC), but a paradigm for prospective confirmation is still lacking. Patientdatasets were mainly downloaded from TCGA, CPTAC and GEO. We conducted downstream analysis by collecting metagenes and generated 42-gene subtype classifiers to elucidate biological pathways. Subsequently, scRNA, eRNA, methylation, mutation, and copy number variation were depicted from a phenotype perspective. Enhancing the clinical translatability of molecular subtypes, preclinical models including CMAP, CCLE, and GDSC were utilized for drug repositioning. Importantly, we verified the presence of previously described three phenotypes including bronchioid, neuroendocrine, and squamoid. Poor prognosis was seen in squamoid and neuroendocrine clusters for treatment-naive and immunotherapy populations. The neuroendocrine cluster was dominated by STK11 mutations and 14q13.3 amplifications, whose related methylated loci are predictive of immunotherapy. And the greatest therapeutic potential lies in the bronchioid cluster. We further estimated the relative cell abundance of the tumor microenvironment (TME), specific cell types could be reflected among three clusters. Meanwhile, the higher portion of immune cell infiltration belonged to bronchioid and squamoid, not the neuroendocrine cluster. In drug repositioning, MEK inhibitors resisted bronchioid but were squamoid-sensitive. To conceptually validate compounds/targets, we employed RNA-seq and CCK-8/western blot assays. Our results indicated that dinaciclib and alvocidib exhibited similar activity and sensitivity in the neuroendocrine cluster. Also, a lineage factor named KLF5 recognized by inferred transcriptional factors activity could be suppressed by verteporfin.

Introduction

Lung cancer is a heterogeneous disease that histologically consists of mainly adenocarcinoma, squamous cell carcinoma, and small cell carcinoma. Among NSCLC, adenocarcinoma is the most predominant subtype and tends to harbor driver mutations, thereby benefiting from targeted therapies. Meanwhile, immune checkpoint inhibitor (ICI) therapies, centering on anti-programmed death 1 (PD-1/PD-L1), become the keystone of first-line therapy for driver-negative NSCLC. Histologic transformation is thought to be a molecular mechanism of therapeutic resistance, and spontaneous transformation (for example, transformation from small cell lung carcinoma to squamous cell lung carcinoma) has also been described in case reports (Davies et al., 2023; Shiba-Ishii et al., 2024). However, there are no approved therapies for cancer lineage plasticity. Existing clinical evidence suggested that the addition of anti-PD-1/PD-L1 therapies improved prognosis in patients treated with combined pemetrexed and platinum and showed great potential value in lung adenocarcinoma (LUAD). The TME suitable for ICI treatments is defined as “hot-immune” with higher cytotoxic T cell infiltration and tumor mutation burden (TMB). In addition, exploratory markers help move toward precision immuno-oncology, i.e., genomics, radiomics, circulating tumor DNA, microbiota in CheckMate-078 trial and MHC-II in ORIENT-11 trial (Pabst et al., 2023; Sun et al., 2023).

The Cancer Genome Atlas (TCGA) revealed the existence of three transcriptional subtypes in LUAD, including bronchioid/terminal respiratory unit (TRU), magnoid/proximal-proliferative (PP), and squamoid/proximal-inflammatory (PI) (The Cancer Genome Atlas Research Network, 2014). TCGA group proved TRU subtype was associated with a better prognosis and was enriched for EGFR mutations. The loss function of KRAS and STK11 occurred frequently in the PP subtype. While the PI subtype was described as a co-occurrence of TP53 and NF1. Importantly, PP and PI patients have a poorer prognosis compared to TRU, which may be due to their highly proliferative characteristics (The Cancer Genome Atlas Research Network, 2014; Ringnér, Jönsson & Staaf, 2016). Ringnér, Jönsson & Staaf (2016) reported a technical bias in the TCGA subtype and that metagenes may be a valid improvement. Moreover, we think it is counterintuitive because lung cancer lineage infidelity should be less proportional (Ferone et al., 2020). To explain this, the literature has outlined convergent pathways about histological variation: neuroendocrine and squamoid phenotypes (Davies et al., 2023).

Recent studies have shown that the transcriptome determines the fate of lung cancer lineage rather than the genome (Tang et al., 2021). In fact, lung cancer lineage has received attention in preclinical models (Suzuki et al., 2019). We suppose lung cancer lineage infidelity may acquire therapeutic resistance. In this study, we investigated the relationship between lung cancer lineage, immunity, and drug utilization. Additionally, portions of this text were previously published as part of a preprint (https://www.biorxiv.org/content/10.1101/2023.01.18.524544v1.full).

Materials and Methods

Clinical cohorts and preprocessing

For the main analysis, a total of 604 patients with stage IB-IIIA LUAD in the TCGA-LUAD (n = 320), GSE72094 (n = 207), and Clinical Proteomic Tumor Analysis Consortium (CPTAC)-LUAD (n = 77) cohorts were included. UCSC Xena website was used to download TCGA level three RNA-seq data (Illumina HiSeq 2000) (The Cancer Genome Atlas Research Network, 2014; Cai et al., 2019; Gillette et al., 2020; Goldman et al., 2020). RNA-seq count data were transformed into Transcripts Per Million (TPM) for analysis. The expression datasets were downloaded from the Gene Expression Omnibus (GEO) database (https://www.ncbi.nlm.nih.gov/geo/) and quantile-normalized. Then transformed using log(x+1) and log2 for TCGA and GEO respectively. Given that the algorithm employed in this study demanded that the matrix be non-negative, this might distort the extremely low expression variables when utilizing log(x+1).

Next, updated clinical information, copy number variations (CNVs), and mutational data for patients between TCGA-LUAD and CPTAC-LUAD were obtained from cBioportal (https://www.cbioportal.org/) (Cerami et al., 2012). High-level focal CNVs including deep amplification and homozygous deletion. For mutational analysis, germline mutations are removed, and “silent/synonymous” is considered as wild-type, i.e., non-mutated. Meanwhile, the cases with the deficiency of tumor staging and overall survival (OS) were excluded. We deleted three patients in TCGA-LUAD who received neoadjuvant chemotherapy before surgery. Please note that detailed information about the patients in this study is shown in Tables S1 and S2.

Transcriptional cluster distribution and pathway analysis

For differentiating clusters, a 42-gene classifier was used to identify three transcriptional clusters in all expression datasets through the non-negative matrix factorization (NMF) algorithm (Table S3) (Brunet et al., 2004). To compare the difference between repeated sampling and single sampling, a hierarchical clustering (HC) heatmap was used as orthogonal verification. For the HC used for single sampling, mean normalized expression and complete distance were adopted. Note that the above methods require a triple class of both gene and sample. A high confidence queue is considered to have an overall accuracy of over 0.8 and Kappa over 0.6 after cross-validation. Additionally, one thousand highly variable gene expression matrix was used as input for principal component analysis using the R package factoextra to assess the dissimilarity of the clusters. To characterize the diversity of clusters, we selected metagenes from previous publications (Table S3) (Sun et al., 2023; Ringnér, Jönsson & Staaf, 2016; Jia et al., 2018). Above scores were calculated on each sample using the R package GSVA (Hänzelmann, Castelo & Guinney, 2013).

Immune cell evaluation

For previously complete immune-related gene sets, gene expression matrices of Pearson’s correlation (Rmin > 0.5) were used to select the streamlined version in treatment-naive and ICIs-treated NSCLC cohorts. The higher correlation to elucidate reproducible patterns of gene expression, the more relevant to the clinic. Finally, effector cells (effector memory CD8+ T and T helper 1 cells), and immunosuppressive cells (Tregs and MDSCs) were retained from the 28 gene sets, which are highly tumor-specific (Table S3) (Jia et al., 2018).

For immune cell infiltration, the CIBERSORT website (http://cibersort.stanford.edu) was used to repeat 1,000 times to assessment for the relative infiltration proportion of 22 types of immune cells (Newman et al., 2015). Patients with a p-value less than 0.05 were retained. Also, tumor purity was evaluated by the ESTIMATE method (Yoshihara et al., 2013).

Genetic perturbation

Methylation data was sourced from the Xena website, and quantified by the β value (0–1) of each CpG site (Goldman et al., 2020). Note that the β value was processed data that was intended to model the degree of methylation, i.e., a quantification of the ratio, and was not the raw signaling value. Then, missing value processing, region annotations, and differential analysis were described separately. The ChAMP R package performed main analyses, utilizing annotation details per Mullen et al. (2020) and Ito et al. (2023) (Tian et al., 2017). The knn method to complement missing values. Furthermore, promoters included three categories: 1stExon, TSS200, and TSS1500, and enhancers were aligned with H3K27ac binding sites. For differential analysis, DeltaBeta, akin to log fold-change, was set at 0.1 and 0.2.

Also, we obtained the expression matrix of enhancer RNA (eRNA) from https://bioinformatics.mdanderson.org/Supplements/Super_Enhancer/5_Super_enhancer_annotation/ (TCGA_RPKM_eRNA_300k_peaks_in_Super_enhancer_LUAD.txt.gz) (Chen & Liang, 2020), and analyzed through two thousand highly variable eRNAs. The database aims to reflect the activity of super-enhancers and quantify them by RNA-seq. The ImmuLnc R package aided in identifying immune-specific eRNA (Li et al., 2020). Moreover, LINE-1 information for patients from Jung et al. (2019) was collected.

Regulatory activities

In summary, we performed regulatory factor activity assessment in single-cell RNA (scRNA) and bulk RNA sequencing as described by the original authors. The SCENIC and dorothea R packages can both be applied to single cells or bulk RNA sequencing (Aibar et al., 2017; Garcia-Alonso et al., 2018). The consistent frameworks proceed in the following three steps: (1) construction of co-expression networks by R package GENIE3 or ARACNe; (2) relationships between transcriptional factors (TFs) and targets, which mainly include genome binding, i.e., predicting motifs; (3) quantifiable enrichment by R packages AUCell or VIPER. Finally, the TFs with high regulatory potential can be inferred by the above analyses.

For scRNA sequencing, we downloaded the GSE148071 raw expression matrix and conducted downstream analysis by the Seurat R package (parameter settings: nFeature_RNA > 200 & nFeature_RNA < 5,000 & percent.mt < 30 & nCount_RNA < 30,000). To infer the activity of TFs in scRNA, the R package SCENIC was performed (Aibar et al., 2017). We then randomly sampled 1,000 cells from GSE148071 and stratified them by the median value of the immune signature. In addition, the flow-sorted epithelial cell profile (EPCAM+ CD45− CD31−) was downloaded and converted into a regulator matrix using the R package dorothea (Garcia-Alonso et al., 2018). Similarly, grouping is done using the epithelial gene set.

Preclinical models utilization

ConnectivityMap (CMAP) database (https://clue.io) which stored predicted compounds perturbation was used (Subramanian et al., 2017). Optional query functions for the CMAP database include gene expression, cell viability, and proteomics. Gene expression and cell viability functions were used in this study, requiring the input of gene signatures and cell line names. The positive value of compounds reflects a consistent trend with the phenotype, but not negative values. To obtain high-quality compounds, we only considered predictions with scores greater than absolute 1.5.

The cell lines RNA sequencing matrix was from the Broad Institute (https://depmap.org/portal/) (Ghandi et al., 2019). Only 127 NSCLC Cancer Cell Line Encyclopedia (CCLE) cell lines labeled as “type-refined==NSCLC” were selected (HCC1588 was excluded because it was associated with COAD). NSCLC UTSW cell lines for validation from McMillan et al. (2018). In addition, patient-derived xenografts (PDXs) RNA data were from the NCI-MATCH trial and Asian cohort GSE78806. All NMF-identified preclinical models including cell lines and PDXs were described in Table S4.

For drug repositioning, CTRP AUC data were from https://ocg.cancer.gov/programs/ctd2/data-portal (Ghandi et al., 2019). Meanwhile, cell lines expression and IC50 matrix from Genomics of Drug Sensitivity in Cancer (GDSC) were downloaded in https://www.cancerrxgene.org/ (Iorio et al., 2016). AUC and IC50 values were defined as a measure of drug sensitivity. We generated per-drug sensitivity scores for each sample via the R package oncoPredict (Maeser, Gruener & Huang, 2021). The oncoPredict R package was used to predict the relative sensitivity of monotherapy based on a batch-corrected expression profile. In addition, we use the manuscript’s orthogonal discovery method, called a random forest, which can effectively capture potential gene profiles regarding drug sensitivity (Rees et al., 2022).

Differential network and gene

Using the limma package of R, differential expression gene (DEG) analysis was performed (Law et al., 2014). DEG was defined as log fold-change >0.7. The R package chNet generated a network (Tu et al., 2021), guided by subtype-specific genes from the CMAP database. Also, hub genes were considered to connect three as well as more genes.

Cell culture and cell viability

The lung adenocarcinoma H1944 cell line was purchased from Pricella, while the BEAS-2B cell line was obtained from the Xinqiao Hospital Cancer Institute. H1944 and BEAS-2B were maintained in RPMI 1640 (SH30809.01 cytiva) supplemented with 10% FBS and 1% streptomycin in 5% CO2 at 37 °C. Cell viability was estimated by CCK-8 assay. BEAS-2B and H1944 cell lines were co-cultured with Dinaciclib (HY-10492; MedChemExpress, Monmouth Junction, NJ, USA) for 48 and 72 h, respectively. Meanwhile, treatment of the H1944 cell line was with Alvocidib (HY-10005; MedChemExpress, Monmouth Junction, NJ, USA) for 72 h.

Western blot analysis

In brief, the H1944 cell line was treated with Verteporfin (HY-B0146; MedChemExpress, Monmouth Junction, NJ, USA) for 96 h. Then, RIPA buffer (P0013B; Beyotime Biotechnology, Haimen, Jiangsu, China) containing 1% PMSF (ST2573; Beyotime Biotechnology, Haimen, Jiangsu, China) was added to the cell culture for lysis. Cell lysates were collected as supernatants after centrifugation and the total protein content was determined by the BCA method. Equal proteins (20 μg/lane) were loaded on SDS-PAGE gels and then transferred to pvdf membranes. After 1 h of closure with Western closure solution (BL535A; Biosharp, Hefei, Anhui, China), pvdf membranes incubate with the primary antibody overnight at 4 °C. Next, the membrane was incubated with the secondary antibody for 1 h at room temperature after washing with TBST. Finally, the target bands were visualized using a chemiluminescent imaging system (FluorQuant AC600; AcuronBio, Darra, Australia). The antibodies used include rabbit anti-KLF5 polyclonal antibody (Cell Signaling Technology, Danvers, MA, USA), and rabbit anti-β-actin polyclonal antibody (BL005B; Biosharp, Hefei, Anhui, China). All bands were normalized to β-actin and the bands were analyzed by Image J.

RNA sequencing on CDKs inhibitors

The experimental protocol was divided into treatment and control groups, with three replicates in each group. In this study, the NCI-H1944 lung cancer cell line was treated with two CDK inhibitors, namely dinaciclib and alvocidib, for 72 h. The transcriptome was assayed at BGI Genomics Co., Ltd. (Shenzhen, China). Total RNA was extracted using Trizol and RNeasy Micro kit (QIAGEN, GER, Venlo, The Netherlands). For library construction, a process involving oligo-dT priming, reverse transcription and fragmentation was employed. Quality control was ensured through Agilent 2100 bioanalyzer (Thermo Fisher Scientific, MA, USA) and real-time quantitative PCR. Using phi29 (Thermo Fisher Scientific, MA, USA) culminates in the formation of DNA nanoballs for sequencing on the DNBSEQ G400 platform. The final library was specified as mRNA type, and quantified using raw count values. Furthermore, the downstream analysis included nine samples and removed missing values.

Statistical analysis

The R package survminer was used to plot the survival by the Kaplan-Meier analysis. ROC and forest diagrams are used for model evaluation. Heatmaps were based on Z-value normalized gene expression, as previously described. Pearson’s chi-squared test or Fisher’s exact test was applied to compare all proportions. The non-parametric test (Wilcoxon-test or Kruskal-Wallis test) was performed to compare variables between groups. All analytical tests were two-sided. A value of p < 0.05 was considered to be statistically significant and adjusted for Bonferroni testing. All codes used for analyses were written in R software.

Results

Transcriptional expression profile classifies three prognostic clusters

Molecular subtypes may provide additional insights for precision medicine. The sample bias and biological explanation of the TCGA group’s subtype were discussed in Ringnér, Jönsson & Staaf (2016) and Wang et al. (2020). Interestingly, the most pronounced neuroendocrine profile was in the PP-3 subtype, which may reflect the original PP subtype, as consistent with previous descriptions (Table S5) (Davies et al., 2023; Shiba-Ishii et al., 2024). In theory, the NMF algorithm can decompose the expression matrix into small metagenes (Brunet et al., 2004). Using the NMF method, a cohort of three hundred and twenty surgically resected stage IB–IIIA tumors from TCGA-LUAD was designed with three clusters after overall consideration (Optimal number of cluster: k = 3, Figs. S1A, S1B). The 42-gene classified patients into bronchioid, neuroendocrine and squamoid clusters, through the NMF method for evaluation (Fig. S1C). In the TCGA-LUAD cohort, our clusters had moderate to high overlap with previous studies, but had significant discrepancies in the Soltis’s cohort (Figs. S2A–S2C, Table S2) (The Cancer Genome Atlas Research Network, 2014; Daemen et al., 2021; Soltis et al., 2022). This may be the biological significance behind the different clusters, determined by molecular classifier or clustering method. For instance, the PP and PI subtypes showed a similar distribution of low NKX2-1 expression compared to the squamoid subtype, representing “NKX2-1 negative” populations (Cardnell et al., 2015). Although three histological lineages of lung cancer are widely accepted, we believe it is necessary to consider the cohort’s inclusion criteria. Thus, an orthogonal approach called HC is used to obtain three high-confidence queues including CPTAC-LUAD, GSE72094, and TCGA-LUAD (see Methods) (Table S6).

Furthermore, we first explore prognostic value among three clusters, the shortest OS was in the squamoid cluster (Figs. 1A, 1B). Then univariate analyses show that age, tumor staging, and squamoid cluster were significantly prognostic (Table S7A). Even if adding age, tumor staging, and gender, squamoid is still an independent prognostic factor using multivariate Cox analyses. Meanwhile, we found a moderate association of clusters with clinical characteristics (Table S7B). After we demonstrated the prognostic significance of the cluster, we asked whether there were significant differences between clusters. Principal component analysis showed that three acquired clusters could be divided with highly variable gene expression data (see Methods) (Figs. 1A, 1B). Among the three clusters, we noticed the squamoid cluster was highly overlapping with patients with lower expression of NKX2-1 (over 90%, Figs. S1D, S1E). The poor prognosis of the squamoid cluster may be related to the inactivation of NKX2-1, the lineage factor in LUAD (Cardnell et al., 2015). Moreover, clusters could distinguish the different lineage fates among three cohorts (Table S7C), supported by previous descriptions (Ferone et al., 2020). In addition, proposed clusters overcome the challenges of RNA-Protein concordance, and are confirmed in samples and PDXs (Tables S2, S4).

Figure 1 Prognosis and predictive chemotherapy information of transcriptional clusters.

Kaplan-Meier Survival is showing prognostic significance by Log-rank test and PCA analysis plotting patients were clustered into three distinct clusters for (A) TCGA-LUAD cohort (n = 320), and (B) GSE72094 cohort (n = 207) (left: survival analysis; right: PCA plotting, the X, Y and Z axes represent the three principal components). (C) Squamoid cluster received cisplatin or not receiving cisplatin from TCGA-LUAD (n = 50) (D) GSE19188 (n = 70) predicted for pemetrexed therapy. (E) Neuroendocrine cluster treated with immune checkpoint inhibitors from SU2C-MARK LUAD (n = 44). Inclusion of only samples within 300 days of overall survival. Legend was labeled in blue (bronchioid), yellow (neuroendocrine), red (squamoid), purple (non-treated), black (treated), and fuchsia (non-neuroendocrine).

To validate the clinical value of clusters, we evaluated cluster-related therapies. Initially, the squamoid cluster in the neoadjuvant population showed improved prognosis post cisplatin-based therapies (Fig. 1C). Additionally, pemetrexed displayed potential efficacy within the bronchioid cluster (Fig. 1D). Robust three clusters were replicated in the SU2C-MARK LUAD cohort, with bronchial and squamoid clusters consistently exhibiting favorable trends (median days: bronchioid: 560; neuroendocrine: 443; squamoid: 547, p = 0.37) (Ravi et al., 2023). Furthermore, the neuroendocrine cluster had the worst OS when considering only individuals treated with ICI for less than one year (Fig. 1E). Nonetheless, there were no discernible differences in progression-free survival across clusters.

In summary, the molecular classifiers trained by the NMF method, can determine the three histologically relevant clusters with prognosis. Highly convincing queues are obtained via NMF and HC algorithms, leading to conservative downstream results.

Comprehensive characterization among three clusters

We selected four lung-specific metagenes to represent clusters described in Table S3 (Sun et al., 2023; Ringnér, Jönsson & Staaf, 2016; Jia et al., 2018). Using the R package GSVA, we found that neuroendocrine and squamoid shared proliferation among three cohorts (Fig. 2A) (Hänzelmann, Castelo & Guinney, 2013). Although metagene patterns were shared in different clusters, the metagenes of surfactant, neurodevelopment, and basal distinctly corresponded to bronchioid, neuroendocrine, and squamoid, respectively, suggesting that metagenes can reflect approximate transdifferentiation directions among clusters. We hoped to understand the immune profiles in transdifferentiation-related clusters. Among three cohorts, the least immune infiltrate was in the neuroendocrine cluster (Fig. 2A). The bronchioid cluster had the highest proportion of resting mast cells, but the lowest proportion of activated memory CD4+ T cells using the CIBERSORT algorithm (Fig. 2B) (Newman et al., 2015). After considering the high-resolution dataset, we found squamoid cluster was most related to T-cell status but had the least CellPhoneDB inferred cellular communication using the Scissor method (Figs. S3A–S3C, Table S8) (Efremova et al., 2020; Sun et al., 2022).

Figure 2 Pathways, immune cells, copy number aberrations and mutations among three clusters.

(A) Heatmap drawing Gene Set Variance Analysis (GSVA) scores for each patient in the CPTAC-LUAD cohort (n = 77), GSE72094 cohort (n = 207) and TCGA-LUAD cohort (n = 320). Z-value GSVA score projected into (−2;2). (B) Box plots exhibiting proportion of activated memory CD4+ T and resting mast immune cells using Kruskal-Wallis test in the CPTAC-LUAD cohort (n = 77), GSE72094 cohort (n = 207) and TCGA-LUAD cohort (n = 320). Asterisks (*, ** and ****) represent p < 0.05, p < 0.01, p < 0.0001, respectively. (C) CDKN2A homozygous deletion and 14q13 high-level amplifications among three clusters in the TCGA-LUAD cohort. (D) Genome plot showing focal chromosomal alterations of neuroendocrine in the TCGA-LUAD cohort. Given the G-score (x axis) for each focus event (y axis). Note that a high G-score means a high probability of occurring events. (E) Line graph showing the percentage distribution of the four major mutations (EGFR, KRAS, STK11 and TP53) among three clusters in the CPTAC-LUAD, GSE72094 and TCGA-LUAD cohorts. (F) Kaplan-Meier plot showing survival time of ZNF536, PXDNL, ADGRB3 and ADGRL3 mutation status among three clusters in the TCGA-LUAD cohort (left: bronchioid; middle: neuroendocrine; right: squamoid).

Next, we examined the CNVs among three clusters, 14q13.3 amplification was the sole region that passed (chi-squared test p < 0.05) based on the cBioportal website and focal-level identification (GISTIC2.0 software parameter: 0.99 confidence). Many 14q13.3 co-alternations (e.g., NKX2-1 and MBIP) were enriched in the neuroendocrine cluster, and the amplification variation might explain the high expression of NKX2-1 (Kwei et al., 2008). In the CPTAC-LUAD cohort, the neuroendocrine cluster also had the highest MBIP alternations although it did not reach corrected statistical significance. Previous studies have proven that inactivation of NKX2-1 would induce squamous differentiation, and CNV data supported this notion, i.e., lower frequencies in squamous cell lung carcinoma compared to LUAD (Davies et al., 2023; The Cancer Genome Atlas Research Network, 2014; Ferone et al., 2020). In line with this, the squamoid subtype had the highest frequency of CDKN2A variations but the lowest 14q13 (Fig. 2C). Further examination revealed that the neuroendocrine cluster had a high frequency of chromosome 8q variation, especially 14q13.3 using the R package maftools (Fig. 2D) (Mayakonda et al., 2018). Furthermore, somatic mutational analysis was implemented, where synonymous mutations were considered as wild. We focused on four major mutations (EGFR, KRAS, STK11, and TP53) in CPTAC-LUAD, GSE72094 and TCGA-LUAD cohorts (Fig. 2E). Importantly, STK11 mutations were mainly distributed in the neuroendocrine cluster. The distribution of mutations among clusters is very different across three cohorts (e.g., KRAS and EGFR), which may be due to ethnicity and smoking history. Then, mutations with high frequency and prognostic significance were listed on the cBioportal website. ZNF536, PXDNL, ADGRB3, and ADGRL3 mutations were almost greater than 10% in frequency, except for a cohort of predominantly non-smokers (Fig. S4A). Our results found that the prognosis of mutations in different clusters could be opposite using the TCGA-LUAD cohort (Fig. 2F). In addition to the known STK11, there were additional mutations related to ICI therapies. Both ZNF536 and ADGRB3 mutations showed favorable prognostic significance, especially ZNF536 mutations (Figs. S4B, S4C). Overall, genomic results are presented among clusters, and the neuroendocrine cluster has multiple genomic vulnerabilities, such as STK11 mutations and NKX2-1 amplifications.

Genetic differences within molecular subtype and related to immunity

Then, DNA methylation and eRNA profiles were depicted in clusters. Enrichment analysis of genes associated with differentially methylated promoter regions revealed bronchioid and neuroendocrine clusters showing dysregulated differentiation and synapses, respectively (Fig. 3A), whereas the squamoid cluster contained an insufficient gene count for enrichment analysis. Subsequently, we explored differentially methylated promoter and enhancer sites with fold changes exceeding 0.1 and 0.2. Across up-regulated loci in subtypes, there were slightly fewer putative promoter sites overall compared to enhancer sites when considering higher fold changes (Figs. 3B, 3C). We found that the neuroendocrine cluster had the least number of specific sites either promoters or enhancers possibly reflecting global methylation levels. Interestingly, the three immune-related profiles including LINE-1, MHC-II enhancers and immuno-specific eRNAs consistently exhibited a decreasing trend from high to low across the bronchial, squamoid and neuroendocrine clusters (Fig. 3D).

Figure 3 Subtypes difference in methylated and eRNA perturbation.

(A) Metascape enrichment (https://metascape.org/gp/) of methylated region-associated genes in bronchioid and neuroendocrine clusters, respectively. (B and C) The number of differentially methylated promoters and enhancers between clusters is shown. The deltaBeta values greater than 0.1 and 0.2 were used as thresholds for differential methylation. (D) Box plots showing the levels of methylation from global LINE-1, MHC-II enhancers and immunospecific super-enhancers in the TCGA-LUAD cohort. Comparison between groups using Wilcoxon test. Asterisks (*, ** and ***) represent p < 0.05, p < 0.01, p < 0.001, p < 0.0001, respectively. (E) Hazard forest plots are generated based on nine enhancer loci that are up-regulated in the neuroendocrine cluster. Note that the abbreviations used in the figure are ‘BR’ for bronchioid, ‘NE’ for neuroendocrine, and ‘SQ’ for squamoid cluster.

Combined with our results, the neuroendocrine cluster appeared to exhibit hypomethylation patterns, and presented resistance to immunotherapy (Ito et al., 2023; Jung et al., 2019). Consequently, we chose to focus on developing methylation sites targeting the neuroendocrine subtype. Given that enhancers exert long-range regulatory effects, loci that were upregulated in neuroendocrine subtypes underwent further screening across two datasets with anti-PD-1 therapy. In the first data named GSE126043, only the AUC values of 1 were included in the analysis (Table S9). Nine highly sensitive diagnostic markers were selected for modeling, with cg03873220 (PARP12) emerging as a potential main effect in GSE119144 (Fig. 3E). While the median value of the probe lacked prognostic significance, three interacting pairs (cg03873220:cg10472651, cg03873220:cg03616827, cg03873220:cg27640794) displayed significance (Table S9). Our exploration has unraveled genetic perturbations within subtypes and identified pertinent features concerning immunotherapy.

Transcriptional clusters design personalized targeted therapy

To facilitate drug utilization and development, the CMAP database is used (see Methods) (Table S10). Despite no common drugs being identified, the bronchioid cluster showed resistance to multiple MEK inhibitors, which may contradict the findings reported by Daemen et al. (2021) (Fig. S5A). Using R package oncoPredict, we further analyzed the CCLE and GDSC databases, and for GDSC only the top 100 compounds with high lethality were included (Table S10). Our results show that MEK inhibitors may favor squamoid cluster based on CCLE and GDSC datasets (Figs. S5B, S5C). Then, subtype-associated differential targets were focused on, using co-expression network construction and direct comparison, respectively. The subtype-associated differential network was constructed based on CMAP experimental perturbations, and experimental evidence supported that TNFRSF12A, which is one of the hub genes in the GSE72094 squamoid network (Fig. S6, Table 1) (Tu et al., 2021; Subramanian et al., 2017). Through differential gene analysis directly, DEGs of subtypes may be druggable targets, such as SLC34A2 (Table S10) (Li et al., 2023).

Table 1 Differential network nodes according to subtype.

Bronchoiod	AKT3, ANAPC5, CIRBP, EGFR, EPCAM, PHGDH, SIAH1	
Neuroendocrine	IFIH1, ZC3H12C	
Squamoid	ADCY3, AKAP17A, ANKRD44, ARHGEF5, COL1A2, COQ6, EZH2, ERMAP, DNMT3A, GNAI2, GYPE, IDH2, IFIH1, IL18, KDELR3, MAP4K4, MCCC1, MET, MYD88, NDUFA9, NKX2-1, PHGDH, PIPOX, RAP1A, RASSF5, SEMA7A, SERPINB5, SLC5A3, TLR2, TNFRSF12A, UNC13B	
Note:

Hubs are thought to connect three as well as more genes.

Based on drug sensitivity data and cell line experiments, our results suggest that alvocidib and dinaciclib are promising in the neuroendocrine cluster and that we should be aware of concentration (Figs. 4A–4C). The second-generation pan-CDK inhibitor dinaciclib may be superior to the first-generation pan-CDK inhibitor alvocidib, our results showed that dinaciclib exerted antiproliferative effect in tumor and normal epithelial cell lines (Fig. 4C). Both dinaciclib and alvocidib reduced interferon and metabolic processes, and increase extracellular matrix and squamoid remodeling (Fig. 4D, Tables S11, S12). Taken together, our results validate for the first time that dinaciclib and alvocidib are highly similar (model prediction: about 50%; transcriptome results: alvocidib: 79%, dinaciclib: 43%, Tables S11, S13).

Figure 4 Dinaciclib and alvocidib predicted from bioinformatics and validated in vitro.

(A) CCLE predicted sensitivity values in clusters using Wilcoxon test. Asterisks (****) represent p < 0.0001 (left: dinaciclib; right: alvocidib). Higher values mean therapy resistance. (B) Cell viability of the NCI-H1944 cell line treated with alvocidib. (C) Cell viability of the NCI-H1944 and BEAS-2B cell lines treated with dinaciclib (left: NCI-H1944; right: BEAS-2B. The horizontal and vertical axes are the concentration and cell survival ratio, respectively, while the IC50 values have been labeled). (D) GO enrichment analysis of NCI-H1944 after treatment with dinaciclib and alvocidib, respectively (left: down-regulated shared pathways; right: up-regulated shared pathways).

Activities infer KLF5 as a driver of lineage development and immune invasion

Using the NSCLC datasets stored at the LCE website (n = 5589, Table S1), the pre-existing TME was simplified into effector and suppressor cells (Cai et al., 2019; Jia et al., 2018). We obtained robust immune gene sets by correlation analysis in individual processed data. Together with our results, the abundance of Treg cells is also a potential indicator of anti-PD-1/PD-L1 therapies (Fig. 5A) (Pabst et al., 2023).

Figure 5 KLF5 as a targetable target inferred from single cells and bulk RNA sequencing.

(A) Heatmap plotting the relative immune infiltration based on Gene Set Variance Analysis (GSVA) scores in Pender’s lung (n = 25), GSE126044 (n = 16) and GSE135222 (n = 27) cohorts (left: Pender’s lung; middle: GSE126044; right: GSE135222, Z-value score projected into (0;1)). (B and C) Transcriptional factors regulation grouped by median signature score in GSE148071 and GSE111907, respectively (GSE148071: 0 means no activity, 1 the opposite; GSE111907: Z-value score projected into (−1;1)). (D) Cell viability of the NCI-H1944 cell line treated with Verteporfin. (E) Verteporfin-induce alterations in the protein level of KLF5 in the NCI-H1944 cell line. The western blot analysis-derived bands were normalized to β-actin. Asterisks (**) represent p < 0.01.

We assume that lung lineage development maintains immune balance. Two networks were constructed based on scRNA and epithelial bulk sequencing, using the immune and epithelial gene sets, respectively (Figs. 5B, 5C) (Jia et al., 2018; Bischoff et al., 2021). Lower immune infiltration may be associated with aggressive tumor progression. Our results suggested that KLF5 inhibited immune activation and bronchial differentiation. Given that KLF5 may be recruited when YAP1 activation, we examined whether verteporfin could inhibit KLF5 (Gokey et al., 2021). As expected, we detected an IC50 value of 1.614 μM for verteporfin in the H1944 cell line using the CCK8 assay, and subsequently treated the cells at the concentration of the IC50 for 96 h (Fig. 5D). Proteins from H1944 cells were then extracted for Western blot assay and found that verteporfin inhibited protein levels of KLF5 (Fig. 5E). We speculate that verteporfin regulates the TME partly through the lineage factor KLF5. Future studies to determine whether verteporfin resolves KLF5-dependence would be interesting.

Clinical translation of transcriptional clusters

Furthermore, common clinical indexes related to immunotherapy including TMB, CNVs, and tumor purity were assessed (Davies et al., 2023; Shiba-Ishii et al., 2024). Among the three clusters, the neuroendocrine cluster had the highest TMB, CNVs, and tumor purity (Figs. S7A, S7B). Indeed, our cluster could also complement the existing immunophenotypes (Thorsson et al., 2018). We observed the proportion of immunophenotypes finding large inconsistencies in different datasets (e.g., TGF beta dominant). Nonetheless, the relative distribution between clusters and immunophenotypes was evident, as reflected in the bronchioid cluster, mainly in lymphocyte depleted (Fig. S8). Overall, transcriptional clusters could provide additional information for immunotherapy, with NKX2-1 and MHC-II as examples (Galland et al., 2021).

Discussion

In this study, we established three clusters across samples, PDXs, and cell lines. Prognosis, genomics, genetics, and TME were depicted among bronchial, neuroendocrine and squamoid clusters. We proved that immune pathways did recognize lineage factors, and focused on the analysis of drug sensitivity differences in three lung cancer lineages.

Our results confirmed that differentiation pathways can profile molecular characteristics of clusters. The bronchioid cluster had high surfactant levels (e.g., NAPSA and SFTPC, etc.), and the neuroendocrine cluster showed neurodevelopment (e.g., ASCL1 and INSM1, etc.), and high expression of keratin genes (e.g., KRT6A and KRT16, etc.) was shown in the squamoid cluster. Among the three clusters, the bronchioid cluster is associated with non-smokers and females, while neuroendocrine and squamoid clusters in contrary. Recently, Roh et al. (2022) depicted vulnerabilities in the PI subtype, whereas we tended to refine precise the PP subtype (Wang et al., 2020). The neuroendocrine cluster was defined as having the highest frequency of STK11 mutations and NKX2-1 amplifications, but the lowest immune infiltration. It is confirmed that bronchioid is immune activated, which may be due to EGFR mutations neglected in the past (Pabst et al., 2023). Indeed, the role of NKX2-1 still be contradictory in immunotherapy, with NKX2-1 positive individuals having higher MHC-II (about 63% overlap in positive immunohistochemistry), while immunosuppressive TME in negative NKX2-1 patients. We speculate that the squamoid cluster may have the worst prognosis due to the hybrid or epithelial-mesenchymal transition state (Li et al., 2022).

From the therapeutic aspect, the squamoid cluster may benefit from the MEK inhibitors after large-scale drug sensitivity analysis. Recently described dysregulation of CDKs in the neuroendocrine cluster, we further show that patients with high proliferation but low squamoid and extracellular matrix remodeling may benefit from pan-CDK inhibitors (Soltis et al., 2022; Roh et al., 2022). Similar functional mechanisms could be identified through expression and drug sensitivity profiling (Rees et al., 2022). Not just CDKs inhibitors, PLK1 inhibitors and EGFR inhibitors also show a high degree of similarity (about 20–50%). Another concept is network-based drug repositioning, we constructed an immunotherapy-related scRNA regulatory network and recognized potential regulators by epithelial bulk sequencing. The above two regulatory networks identified a lineage factor named KLF5, and its dependence on the “NKX2-1 negative” population remains unclear. Investigating cooperation between lineage factors has become popular, and the association has been reported between NKX2-1, KLF5 and YAP1 (Gokey et al., 2021). This study found that verteporfin (a YAP signaling antagonist) did inhibit the level of KLF5.

Emerging evidence in NSCLC research suggests that scRNA analysis was good at resolving heterogeneity, but had difficulty in simultaneously identifying cancerous and non-cancerous components (Li et al., 2022; Salcher et al., 2022). In this study, we implemented two alternative strategies, one focusing on scRNA integration bulk and the other on flow-sorted epithelial profile. Li et al. (2022) showed co-expression of lineage, however, bronchioid became overwhelmingly dominant and did not determine the lineage classification of patients. In contrast, the transcriptional clusters allow for personalized therapies in identifiable lung cancer lineage (Ferone et al., 2020; Tang et al., 2021; Suzuki et al., 2019). Additionally, we compare correlation analysis of global, lineage-specific, our 42-classifier and immune pathways genes, separately (Figs. S9A, S9B). Lower correlations from cell lines were observed considering lineage and our 42-classifier. Moreover, about 86% of lineage-specific genes had a low expression in McMillan et al. (2018). Our results show that partly lineage features of the cell lines were lost, thus the validity of preclinical models needs to be considered (Ferone et al., 2020). We need to interpret this result with caution, due to the lack of morphological confirmation.

The direction of future research is the combination of the genetic and non-genetic, and arguably epistatic oncogenic transcriptomic landscape. The important questions are whether there is a need to intervene in transdifferentiation and how to rely on the lineage to target therapeutic vulnerabilities. To date, the transformations toward neuroendocrine and squamoid phenotypes have emerged as potential mechanisms of resistance to tyrosine kinase inhibitors therapy in LUAD, e.g., EGFR inhibitors. A recent study has also shown that a similar mutational profile was found in the neuroendocrine lineage (Sivakumar et al., 2023). Drawing on our results may help to find shareable targets in lung cancer.

Twenty years ago, it was reported that neuroendocrine and squamoid phenotypes were in a subset of LUAD (Bhattacharjee et al., 2001; Garber et al., 2001). We verified that the above phenotypes could be reflected in LUAD by RNA-seq and microarrays, which was a prerequisite for the identification of shareable targets. In other words, our study has the significant advantage of modeling lineage fate compared to previous ones (The Cancer Genome Atlas Research Network, 2014; Gillette et al., 2020; Soltis et al., 2022; Roh et al., 2022; Bhattacharjee et al., 2001; Garber et al., 2001). The phenotype-drug associations were confirmed, i.e., some drugs may induce differentiation pathways, such as dinaciclib and alvocidib.

Although high heterogeneity in advanced stage tumors, unperturbed labels of DEGs derived from subtypes are sufficient to guide clinical trials, such as SU2C-MARK and Piedmont. Furthermore, some of the DEGs have been trained as classifiers and validated with the NanoString platform (Karlsson et al., 2019). We emphasized that molecular markers described in the literature are within our DEG subset, like CHGB which is upregulated in the neuroendocrine group, and TRIM29 which is upregulated in the squamoid group (Davies et al., 2023; Shiba-Ishii et al., 2024). Whereas the bronchioid group is characterized by pulmonary diseases (such as TMPRSS2 and DPP4), and activated cell-surface proteins (Table S10).

Clinical factors (e.g., smoking) in LUAD, lead to confusion in the interpretability of molecular characteristics. We observed all neuroendocrine cluster belongs be smokers in the CHOICE cohort, thus the absence in the OncoSG cohort is explainable (Table S2) (Zhang et al., 2019; Chen et al., 2020). To our knowledge, the TRU and non-TRU binary classification may apply to cohorts with a high proportion of non-smokers (Karlsson et al., 2019; Liljedahl et al., 2021). Finally, there is still a need to determine the identity and prognosis of neuroendocrine characteristics.

A significant limitation is our primary analysis of existing anticancer compounds within the database, overlooking untapped pharmaceutical resources like traditional Chinese medicine. Despite these shortcomings, we propose RNA-Protein consistent clusters and focus on transcriptome-based drug repositioning, and in the future, we will also integrate chromatin accessibility and radiomics. Also, the implementation of metagenes via the NanoString platform is valuable.

Conclusions

We propose three transcriptomic clusters including bronchioid, neuroendocrine and squamoid phenotypes to mimic histologic fate in lung cancer. In this regard, we confirmed the link between phenotype and drug through evidence of molecular subtypes, regulatory networks, and lab experiments. Genomics, genetics, and clinical features of lung adenocarcinoma are profiled with an emphasis on clinical translation. Chemotherapy, immunotherapy, and targeted therapies could have differences in subtypes, and the bronchioid phenotype has the greatest treatment potential. In lung adenocarcinoma, the neuroendocrine phenotype is resistant to immunotherapy but may benefit from CDKs inhibitors. Among the three phenotypes, the worst prognosis is in the squamoid, where lineage infidelity plays a decisive role, probably due to the inactivation of NKX2-1. Finally, lineage molecules may be involved in both epithelial and immune regulation, like KLF5. Lineage-based concepts promise to guide drug discovery, especially shareable targets.

Supplemental Information

Supplemental Information 1 Code integrated.

Supplemental Information 2 Supplementary Figures.

Supplemental Information 3 Supplementary Tables.

Supplemental Information 4 Original cell line experimental data.

1. CCK8 assay 2. Western-blotting ploting KLF5 protein change

We thank BGI Genomics Co., Ltd. (Shenzhen, China) for transcriptional sequencing, and appreciate the data from the TCGA, CPTAC and GEO datasets.

Abbreviations

NSCLC Non-small cell lung cancer

TME Tumor microenvironment

ICI Immune checkpoint inhibitor

PD-1/PD-L1 Programmed death 1

LUAD Lung adenocarcinoma

TMB Tumor mutation burden

TCGA The Cancer Genome Atlas

TRU Terminal respiratory unit

PP Proximal-proliferative

PI Proximal-inflammatory

CPTAC Clinical Proteomic Tumor Analysis Consortium

TPM Transcripts Per Million

GEO Gene Expression Omnibus

CNVs Copy number variations

OS Overall survival

NMF Non-negative matrix factorization

HC Hierarchical clustering

eRNA Enhancer RNA

scRNA Single-cell RNA

TFs Transcriptional factors

CMAP ConnectivityMap

CCLE Cancer Cell Line Encyclopedia

PDXs Patient-derived xenografts

GDSC Genomics of Drug Sensitivity in Cancer

DEG Differential expression gene

Additional Information and Declarations

Competing Interests

Author Contributions

Data Availability

The authors declare that they have no competing interests.

Longjin Zeng analyzed the data, prepared figures and/or tables, authored or reviewed drafts of the article, and approved the final draft.

Longyao Zhang performed the experiments, prepared figures and/or tables, and approved the final draft.

Lingchen Li performed the experiments, prepared figures and/or tables, and approved the final draft.

Xingyun Liao performed the experiments, prepared figures and/or tables, and approved the final draft.

Chenrui Yin performed the experiments, prepared figures and/or tables, and approved the final draft.

Lincheng Zhang performed the experiments, prepared figures and/or tables, and approved the final draft.

Xiewan Chen performed the experiments, prepared figures and/or tables, and approved the final draft.

Jianguo Sun conceived and designed the experiments, authored or reviewed drafts of the article, and approved the final draft.

The following information was supplied regarding data availability:

The code and the cell line description are available in the Supplemental Files.

The raw data related to small molecule compounds is available at OSF: Zeng, Longjin. 2024. “Lung_lineage.” OSF. September 5. osf.io/rtwsh.

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
