# Peer review of "RNA sequencing identifies lung cancer lineage and facilitates drug repositioning"

_PeerJ, doi:10.7717/peerj.18159_

## Round 0.1 · original submission · Major Revisions

Dear authors,
please refer to the reviewers' comments for details on the necessary revisions and questions to answer. It is ok to require more time than the standard provided. Carefully proofread everything before resubmission to facilitate moving forward. Many thanks.

Reviewer 1 ·

Basic reporting

Researchers utilized gene expression datasets from TCGA and GEO to investigate subtypes of non-small cell lung cancer. Their study confirmed the existence of three distinct lineages: bronchioid, neuroendocrine, and squamoid. They further examined the differences in gene expression, epigenetics, and tumor microenvironment (TME) across these subtypes.
Overall, the manuscript is challenging to read due to various grammatical issues that render many sentences difficult to understand. For example:
About HC visual storage as the python script.
Using a large NSCLC-cohort (n = 5589, Supplementary Table 1), our previous view was refined, and further, dissect the pre-existing TME specifically into effector and suppressor cells.

The manuscript's structure is poorly organized. For instance, the second paragraph of section 3.1 begins with a survival analysis and abruptly transitions to an analysis of NKX2-1 expression.

The font size in many figures is too small to read, and the text appears stretched.

Experimental design

Did the authors perform the RNA-seq experiments used in figure 4D? If so, they should include the protocols in the methods section.

BEAS-2B and H1944 cell lines were co-cultured with Dinaciclib for 48 and 72 hours, respectively. What is the reason for using different culture durations?

The authors should mention the concentration of Verteporfin and indicate if there is a change in cell viability after Verteporfin treatment.

Validity of the findings

"Using the R package GSVA, we found that neuroendocrine and squamoid shared proliferation among four cohorts (Fig. 2A)." However, only three cohorts are shown in figure 2A.

Why do the authors focus on 14q13.3 in the CNV analysis?

In figure 2F, are all types of mutations (missense, nonsense, silent, etc.) included in the mutate group?

The two figures in 4C should have the same x-axis scales for easier comparison.

In figure S1A, the authors chose three clusters according to the TCGA-LUAD dataset. However, in figure S2C, their three clusters differ significantly from the original three clusters proposed in the TCGA-LUAD paper. They should explain the reason for this discrepancy.

Reviewer 2 ·

Basic reporting

The figures in the article suffer from significant issues with clarity and resolution. The poor quality of the figures makes it challenging to interpret the results effectively. Improving the resolution would greatly enhance the clarity and impact of the study.

Experimental design

1. In the Methods section, particularly in the regulatory activity part, it would be beneficial to elaborate on how the cisTarget data was utilized to identify transcription factors (TFs). Providing this detail would clarify the approach taken and enhance the reproducibility of the study. Additionally, when discussing the findings related to KLF5, there is a disconnect in the flow of information.

2. It appears there might be a mistake regarding the R package used for identifying immune-related enhancer RNAs (eRNAs). It seems the correct package utilized by the authors is ImmLnc, not lmmuLncRNA

3. In Figure 4D, it would be beneficial to conduct pathway analysis separately for each treatment group. This approach would provide clearer insights into the specific biological pathways influenced by each treatment instead of only showing common pathways.

Validity of the findings

Overall, the approach appears promising, but improvements in readability and image quality would enhance the interpretability of the results.

Additionally, providing all scripts used in the manuscript would enhance transparency and facilitate the reproducibility of the findings.

Additional comments

In line 232, the manuscript mentions four cohorts, but the analysis appears to involve only three cohorts.

·

Basic reporting

Recent advances have improved survival in non-small cell lung cancer (NSCLC), but a supporting paradigm for prospective confirmation is needed. This study analyzed gene expression datasets and drug sensitivity databases and identified three clusters (bronchioid, neuroendocrine, and squamoid), with specific genetic and immune characteristics, and varying drug sensitivities, notably MEK inhibitors for squamoid and dinaciclib/alvocidib for neuroendocrine clusters. Overall, the manuscript is written in a professional and technically appropriate manner. However, it would benefit from some revisions to improve clarity and conciseness, especially in complex sections.
Major issues with basic reporting are as follows:
1) The figures in the main manuscript suffer from poor resolution and inconsistent font sizes, which significantly impede readability. The low-quality resolution blurs critical details, while the varying font sizes make it difficult to interpret labels and annotations accurately. This hinders the overall clarity and comprehension of the presented data.
2) Code used to generate figures and execute the bioinformatics workflow for quantifying results from the datasets and downstream analyses has not been provided, thereby impeding the reproducibility of the findings.
Minor issue with basic reporting is as follows:
1) Certain sections could benefit from improved clarity and precision in language. For instance, sentences on lines 23, 77, 91, 121, and 128 have phrasing that could be revised to enhance comprehension.
2) The introduction could be more focused on specifying the knowledge gap that this research aims to fill. Although it mentions the lack of a targetable driver in a significant proportion of NSCLC cases, the specific gap that this study addresses could be highlighted earlier and more clearly.
3) A more detailed justification for the study could strengthen the introduction. Specifically, expanding on the potential implications of the findings for clinical practice would provide a more compelling rationale for the research.
4) Lines 41-50: The introduction of the genetic basis of NSCLC and the role of tyrosine kinase inhibitors could be simplified for better understanding.
5) Lines 108-110: The explanation of methylation data and promoter categories could be made more concise and clear.
6) Lines 121-125: The description of the single-cell RNA sequencing analysis and the use of the Seurat R package might benefit from additional explanation.
7) Lines 188-192: The summary of transcriptional expression profiles and their classification into prognostic clusters could be streamlined.
8) Lines 253-257: The discussion of genetic alterations and their implications for immune response could be clarified to ensure readers understand the significance.
9) Lines 290-293: The introduction to personalized targeted therapy and the role of DEGs could be made clearer.

Experimental design

Major issues with experimental design are as follows:
1) The manuscript specifies the R packages used for each of the analyses but does not include the exact functions or parameters for a function or the versions of the packages employed. This omission impedes reproducibility, as different versions of packages may yield varying results, and the absence of function details makes it challenging to replicate each analysis precisely.
2) The manuscript needs to provide a detailed description of how missing data was handled for each of the analyses. This information is crucial for understanding the robustness of the results and ensuring that the methods used to address missing data do not introduce bias or affect the validity of the findings.

Minor issues with experimental design are as follows:
1) While the statistical methods are well-described, providing more detailed information about the adjustment for multiple testing (e.g., Bonferroni correction) and the rationale behind the choice of specific statistical tests would further strengthen the methodology section.
2) The manuscript indicates that different normalization techniques were applied to the TCGA and GEO datasets. The authors need to elaborate on the rationale behind using distinct normalization methods, as this can impact the comparability and interpretation of the results across these datasets. Providing a clear justification will enhance the understanding of the chosen approach and its implications for the analysis.

Validity of the findings

Minor issues with the results and discussion section are as follows:
1) The discussion section sometimes reiterates results without providing additional interpretation or context. It would be more effective to focus on the implications of the findings rather than repeating the data.
2) While the discussion section references relevant literature, it could more thoroughly compare and contrast the study’s findings with previous research to highlight novel contributions and contextualize results within the broader scientific field.
3) While some limitations are mentioned, a more comprehensive and transparent discussion of the study’s limitations, including potential biases and methodological constraints, would improve the integrity and credibility of the work.

Additional comments

The manuscript presents a well-executed study with significant findings relevant to the field of lung cancer research. It is overall strong in terms of English language, experimental design, and findings. However, addressing the above concerns will significantly enhance the clarity, rigor, and impact of the study, ensuring it meets the high standards required for publication.

---

## Round 0.2 · Major Revisions

Dear authors,

Many thanks for your re-submission. At this stage, I would like to know if you are able to revise and/or reply to the questions of reviewer #1 or not? I am also not happy with some answers to the reviewers, such as the answer "X company did this. ... you are responsible for the WHOLE work, so you should obtain from the company what they did. otherwise, how well equipped really are you to judge, evaluate and critically think about the data received?

Avoid "blanket statements", short neutral replies and anything just to try to quiet down the reviewers; incongruousness between manuscript and reply to reviewers are also to be avoided.

Methods and data presentation is pivotal for PeerJ and it must be crystal clear. It also helps if you straight away present figures with EXCELLENT resolution (this includes fonts!); as per the journal's authors guidelines, at least.

Many thanks in advance.

Reviewer 1 ·

Basic reporting

I appreciate the authors' efforts to address the issues I identified during the initial review. However, I still find the manuscript challenging to read. For example, some sentences in the abstract are difficult to understand: “Metagenes included collected gene sets and 42-gene sample classifiers to explain biological pathways and clustering analysis, respectively,” and “Cluster-based drug repositioning through databases including CMAP, CCLE, and GDSC.” Other sentences in the abstract could be improved, such as deleting “and preclinical models” from “Gene expression datasets were downloaded from TCGA, CPTAC, and GEO and preclinical models.” Additionally, “Further analysis is the estimation of the relative cell abundance of the tumor microenvironment (TME)” can be changed to “We further estimated the relative cell abundance of the tumor microenvironment (TME).”
The figure quality is still not suitable for publication. I have to download the original figures and zoom in to read the small fonts in figures 2F, 3D, and 3E.

Experimental design

The answers to some of my previous questions are not satisfactory:
“Did the authors perform the RNA-seq experiments used in figure 4D? If so, they should include the protocols in the methods section.”
The authors added one sentence in section 2.10: “The transcriptome was assayed at BGI Genomics Co., Ltd (Shenzhen, China).” However, it is still unclear which protocol was used. The authors also indicated in their response that this dataset is from a previous manuscript (https://doi.org/10.21037/jtd-23-1164), but they did not cite this paper.
“Why do the authors focus on 14q13.3 in the CNV analysis?”
I understand that CNV plays an essential role in lung cancer progression. The authors should explain why they focus on 14q13.3 CNV rather than CNVs in other regions. It would be helpful if they could add more explanations on how genes within 14q13.3 may contribute to lung cancer

Validity of the findings

no comment

Reviewer 2 ·

Basic reporting

The authors have effectively addressed the reviewer's comments, resulting in improved manuscript readability and enhanced figure quality. However, it would be beneficial to standardize and increase the font size, as the figures currently have varying font sizes, with some being particularly small and hard to read.

Experimental design

No comment

Validity of the findings

No comment

·

Basic reporting

The revised manuscript now exemplifies good basic reporting following comprehensive integration of peer review feedback.The figures have been improved with higher resolution and consistent font sizes, enhancing readability and clarity. The provided R code helps with reproducibility of the findings. Additionally, the manuscript's language has been refined for better clarity and precision, particularly in the specified sections. The introduction has been restructured to more clearly highlight the knowledge gap this research addresses, with a more detailed justification added to emphasize the potential clinical implications. Various sections, including the discussion of NSCLC, genetic alterations, methylation data, and transcriptional profiles, have been simplified and clarified to improve overall comprehension.

Experimental design

All the issues related to the experimental design have been addressed in the revised manuscript. Detailed description of how missing data was handled for each analysis has been included, addressing concerns about the robustness and validity of the results. Furthermore, the methodology section has been strengthened with more detailed information on the adjustment for multiple testing and the rationale behind the choice of specific statistical tests. The manuscript also elaborates on the rationale for using different normalization techniques for the TCGA and GEO datasets, providing a clearer understanding of the chosen approach and its impact on the analysis.

Validity of the findings

The minor issues with the results and discussion section have been addressed in the revised manuscript. The discussion now focuses more on the implications of the findings, rather than reiterating the results. Additionally, the study's findings are more thoroughly compared and contrasted with previous research to highlight novel contributions and better contextualize the results within the broader scientific field.

Additional comments

The revised manuscript presents a well-executed study with significant findings relevant to the field of lung cancer research. It is strong in terms of English language, experimental design, and findings. The previously mentioned concerns have now been addressed, which significantly enhances the clarity, rigor, and impact of the study, ensuring it meets the high standards required for publication.

---

## Round 0.3 · accepted · Accept

Dear authors,
I am happy to let you know that i am accepting your manuscript for publication. In moving forward, production will let you know about minor revisions like punctuation or grammar. Please, be throughout in your proofreading. Also, some figures may need to be divided in two or some other solution to improve clarity and resolution, in order to properly allow readers to visualize everything in the published pdf. Many thanks and congrats!

Reviewer 1 ·

Basic reporting

I appreciate the authors' efforts to add the detailed protocols for RNA-seq method. Although there are still many language issues making some sentences difficult to understand, I'm OK to accept the manuscript if the editor is fine with it.

Experimental design

no comment

Validity of the findings

no comment